# Relationship between Circulating Galectin-3, Systemic Inflammation, and Protein-Energy Wasting in Chronic Hemodialysis Patients

**DOI:** 10.3390/nu13082803

**Published:** 2021-08-16

**Authors:** Ming-Tsun Tsai, Shuo-Ming Ou, Huan-Yuan Chen, Wei-Cheng Tseng, Kuo-Hua Lee, Chih-Yu Yang, Ruey-Bing Yang, Der-Cherng Tarng

**Affiliations:** 1Department of Medicine, Division of Nephrology, Taipei Veterans General Hospital, Taipei 11217, Taiwan; mingtsun74@gmail.com (M.-T.T.); okokyytt@gmail.com (S.-M.O.); wctseng@gmail.com (W.-C.T.); dadabim3520@gmail.com (K.-H.L.); cyyang3@vghtpe.gov.tw (C.-Y.Y.); 2Institute of Clinical Medicine, School of Medicine, National Yang Ming Chiao Tung University, Taipei 11221, Taiwan; 3Center for Intelligent Drug Systems and Smart Bio-Devices (IDS2B), National Yang Ming Chiao Tung University, Hsinchu 30010, Taiwan; 4Institute of Biomedical Sciences, Academia Sinica, Taipei 11529, Taiwan; hchen9@ibms.sinica.edu.tw; 5Department and Institute of Physiology, National Yang Ming Chiao Tung University, Taipei 11221, Taiwan

**Keywords:** galectin-3, hemodialysis, lean tissue mass, protein-energy wasting, systemic inflammation

## Abstract

Galectin-3 reportedly participates in the inflammatory process that causes insulin resistance in the target tissues. However, the role of high plasma galectin-3 levels as an indicator of protein-energy wasting (PEW) in patients undergoing maintenance hemodialysis remains unclear. This study included 240 hemodialysis patients (64.5 [55.3−74.0] years, 35.8% women) from a tertiary medical center. A baseline assessment of demographic and clinical data, biochemical parameters, and body composition was conducted. Plasma galectin-3 and other biomarkers were measured using a multiplex bead-based immunoassay. Participants were then divided into two subgroups depending on the median value of plasma galectin-3. Malnutrition was identified using the geriatric nutritional risk index (GNRI) and the criteria of the International Society of Renal Nutrition and Metabolism. Independent risk factors for elevated plasma galectin-3 and malnutrition were identified by multivariate logistic regression. The high galectin-3 group was more likely to be older, have lower lean tissue mass and GNRI scores, be diagnosed with PEW, dialyze through a tunneled catheter, and have higher circulating IL-6, TNF-α, and MCP-1 concentrations than the low galectin-3 group. After multivariate adjustment, only low mean arterial pressure, dialyzing with tunneled cuffed catheters, and elevated systemic inflammatory markers correlated with high galectin-3 levels. Plasma galectin-3 concentrations also increased significantly in hemodialysis patients with PEW. However, compared with other commonly used nutritional indicators, galectin-3 did not show superiority in predicting PEW. Although the plasma galectin-3 levels correlated with PEW severity, this correlation disappeared after adjustment for potential confounding variables (OR, 1.000; 95% CI, 0.999–1.001). In conclusion, plasma galectin-3 is a valuable biomarker for systemic inflammation but is less prominent for PEW in patients with maintenance hemodialysis. Further identification of novel biomarkers is required to detect patients at risk for malnutrition and implement appropriate interventions.

## 1. Introduction

Chronic kidney disease (CKD) is a major noncommunicable disease with an increasing prevalence over the past few years [1,2]. Most patients with CKD, especially those on long-term dialysis, have persistent low-grade inflammation related to protein-energy wasting (PEW), premature cardiovascular disease (CVD), osteoporosis, and general frailty [3,4,5]. In CKD, factors contributing to chronic inflammation include abnormal mineral metabolism, cellular senescence, intestinal dysbiosis, acid–base imbalance, oxidative stress, and accumulated nonenzymatic glycation [6,7,8]. These factors can initiate and perpetuate uremic inflammation through the mechanisms involved in immune system senescence, innate immunity activation, and inflammatory response dysregulation [7]. Among the inflammation related complications, PEW is characterized by adipose tissue browning, muscle wasting, and resting-energy expenditure increase, ultimately leading to homeostatic imbalance [9]. PEW is by far the leading risk factor associated with poor clinical outcomes in patients with CKD, even after adjusting for comorbidities [10,11,12]. Hence, in many nutrition screening tools, determining the inflammatory status or burden of disease has become the widely accepted approach for diagnosing PEW despite having some controversy [13,14].

Galectin-3 is a β-galactoside-binding protein that is fundamental in a number of cellular functions, including cell adhesion, proliferation, apoptosis, signal transduction, and immune and inflammatory response regulation [15,16]. This protein is highly expressed in different types of human cells, such as immune and inflammatory cells, fibroblasts, endothelium, and epithelium [17]. In addition, galectin-3 has context-dependent effects in renal physiology and pathophysiology [18]. During nephrogenesis, galectin-3 is essential to maintain epithelial polarity via its effect on centrosome biology [19]. Conversely, it is elevated in the serum and renal tissue of patients with kidney disease and involved in tissue inflammation and fibrosis [20]. In patients with impaired renal function, an elevation of circulating galectin-3 is associated with increased risks for incident CKD, progressive renal function loss, and adverse cardiovascular events [21,22,23]. Moreover, galectin-3 might be a robust biomarker for cardiovascular complications and all-cause mortality in patients with non-dialysis-dependent CKD. However, it is less reliable in predicting the prognosis of patients with end-stage renal disease (ESRD) [23,24].

The role of galectin-3 in nutritional status assessment has not been explored until recently. For patients with biliary tract cancer, an increased plasma concentration of galectin-3 is associated with the risk of malnutrition [25]. In addition, the serum galectin-3 levels in patients newly diagnosed with colorectal carcinoma showed a significant inverse correlation with some nutritional parameters, including prealbumin, retinol-binding protein, and transferrin [26]. However, despite the abovementioned findings, the relationship between circulating galectin-3 and PEW presence in patients with CKD remains unclear. Thus, this study determined the plasma galectin-3 levels among patients with maintenance hemodialysis. We aimed to investigate the association between the plasma galectin-3 levels, systemic inflammation, and nutritional status, and evaluate the predictive ability of circulating galectin-3 for PEW in these patients. Our findings would provide useful information for better diagnosis and management of CKD.

## 2. Materials and Methods

### 2.1. Research Design and Patients

Between 1 January and 31 December of 2020 in a tertiary medical center, this cross-sectional study initially recruited 262 adult patients (≥20 years old) on maintenance hemodialysis treatment thrice weekly for at least 12 weeks. Residual renal function, defined as the renal urea clearance greater than 2 mL/min, was lost in all cases. The exclusion criteria were as follows: (1) total hemodialysis treatment time < 12 h per week (*n* = 6); (2) having a prosthetic limb (*n* = 2); (3) having an implanted pacemaker or cardioverter-defibrillator (*n* = 2); and (4) having a history of solid tumors (*n* = 3), bacterial infections (*n* = 5), or hepatobiliary disease (*n* = 4). Ultimately, 240 patients were included. On 5 July 2019, the Institutional Review Board of Taipei Veterans General Hospital approved this study (protocol code: 2019-07-026BC). Furthermore, this study conformed to the Declaration of Helsinki. Written informed consent was obtained before the participants entered the research.

The complete medical history of each patient was recorded and retrospectively assessed. Hypertension was defined as a blood pressure level of ≥140/90 mmHg or the use of antihypertensive agents. Diabetes mellitus was defined as the use of insulin and/or oral antihyperglycemic agents to maintain optimal glucose homeostasis. CVD was defined as having past histories of stroke, peripheral artery disease, or coronary heart disease.

### 2.2. Laboratory Measurements

We collected fasting blood specimens before dialysis initiation on the midweek session. Plasma was separated within 30 min and stored at −80 °C for subsequent analyses. Then, the plasma levels of galectin-3, interleukin-6 (IL-6), tumor necrosis factor-alpha (TNF-α), and monocyte chemoattractant protein-1 (MCP-1/CCL2) were detected using an inhouse multiplex bead-based immunoassay (MBIA) according to a published protocol [27,28]. Briefly, we added the samples and standard dilutions to 96-well microplates containing fluorescence-encoded microspheres precoated with analyte-capturing antibodies. After an incubation period and several times of washing, biotin-labeled secondary antibodies against the target of interest were added, forming an antibody/antigen/antibody sandwich. Subsequently, a streptavidin-phycoerythrin conjugate was added to form the final complex. Plasma biomarker concentrations were then measured using the Bio-Plex^®^ 200 analyzer (Bio-Rad Laboratories, Hercules, CA, USA).

Using a Hitachi 7600 automated chemistry analyzer (Hitachi, Tokyo, Japan), we measured biochemical parameters such as serum albumin, blood urea nitrogen, creatinine, calcium, and phosphate. Dialysis adequacy was determined from the single pool Kt/V for urea (spKt/V_urea_) using the second generation Daugirdas’ equation [29]. The daily protein intake in dialysis patients was calculated using the normalized protein catabolic rate (nPCR) as described previously [30]. Predialysis blood pressure level was obtained in the nonaccess arm after 5 min of seated rest. In calculating the mean arterial pressure (MAP), we added the diastolic blood pressure and one-third of the pulse pressure.

### 2.3. Measurement of Lean Body Mass

The body composition of patients undergoing hemodialysis was evaluated by a multifrequency bioimpedance analysis (MFBIA) device (InBody S10; BioSpace, Seoul, Korea). On the day of blood sampling, patients underwent MFBIA approximately 30 min after a dialysis session. The bioelectrical impedance measurements were obtained using a set of frequencies (1, 5, 50, 250, 500, and 1000 kilohertz) at the five segments of the body (left arm, right arm, trunk, left leg, and right leg). Then, body composition parameters such as fat mass, fat-free mass, and ratio of extracellular water (ECW) to total body water (TBW) were measured using preprogrammed prediction equations provided by the manufacturers and normalized to the square of the individual’s height in metres. We used fat-free mass index (FFMI, kg/m^2^) as a surrogate to estimate lean body mass.

### 2.4. Assessment of Nutritional Status

Malnutrition was identified by the Geriatric Nutrition Risk Index (GNRI) and the International Society of Renal Nutrition and Metabolism (ISRNM) criteria for PEW [31,32]. The GNRI was calculated as follows: GNRI = 1.489 × albumin (g/L) + 41.7 × [dry weight (DW)/ideal body weight (IBW)]. IBW was determined by multiplying an ideal body mass index (BMI) of 22 kg/m^2^ with the person’s height in meters squared. When the patient’s DW exceeded the IBW, the DW/IBW ratio was set to 1. A GNRI score of ≤98 indicated a risk for malnutrition [27,31].

The ISRNM expert panel recommended four main categories, namely, biochemical indicators, body mass, muscle mass, and nutrient consumption, for PEW clinical diagnosis [32]. In this study, the diagnostic criteria of PEW were as shown below: (1) biochemical indicator: serum albumin levels < 3.8 g/dL; (2) body mass: a BMI < 23 kg/m^2^; (3) muscle mass: muscle mass loss > 5% (muscle atrophy) measured by MFBIA on two separate occasions over a 3-month period; and (4) nutrient consumption: nPCR less than 0.8 g/kg/day. When at least three criteria were met, the patient was diagnosed with PEW.

### 2.5. Statistical Analysis

The results are expressed as counts and percentages for the categorical data, and mean ± standard deviation or median and interquartile range for the continuous normally or non-normally distributed variables. The patients were classified into two different groups according to the median value of plasma galectin-3. The two groups were compared using the chi squared (χ^2^) test, Student’s *t*-test, or Mann–Whitney U test, as appropriate. The relationships between the studied variables were explored by Spearman rank correlation. The independent risk factors for elevated plasma galectin-3 and malnutrition were identified using multinomial logistic regression analysis. Age, sex, and variables with *p* < 0.1 in the analysis of univariate data were used in the multivariate analysis. Multicolinearity was identified by calculating the variance inflation factor (VIF) for each predictor in multivariable models, and a VIF ≥ 2.5 indicated considerable collinearity. Furthermore, the diagnostic accuracy of the GNRI, galectin-3, IL-6, TNF-α, and MCP-1 in predicting PEW was evaluated by constructing receiver operating characteristic (ROC) curves. The optimal cutoff point and the maximum summation value of sensitivity and specificity were determined using the Younden index. The areas under the curve (AUC) of each marker were compared using DeLong’s test [33]. In this study, *p* < 0.05 was considered statistically significant. All statistical data were analyzed using SPSS Statistics 23 (SPSS Inc., Chicago, IL, USA) and GraphPad Prism 9 (GraphPad Software Inc., San Diego, CA, USA).

## 3. Results

### 3.1. Clinical Characteristics of Hemodialysis Patients Stratified by the Plasma Galectin-3 Level

Table 1 presents the clinical features of the included patients. The median age was 64.5 years, and 35.8% of them were women. At enrollment, the median dialysis vintage was 49 (29–93.8) months, and the mean spKt/V_urea_ was 1.53 ± 0.26. Hypertension, diabetes, and CVD were found in 212 (88.3%), 134 (55.8%), and 113 (47.1%) subjects, respectively. Around 80% of the participants dialyzed with arteriovenous (AV) fistulas. The median GNRI score was 101.3, and 62 (25.8%) individuals had a low (≤98) GNRI. According to the ISRNM criteria, 48 patients (20.0%) were diagnosed with PEW. Moreover, the median plasma concentrations of galectin-3, IL-6, TNF-α, and MCP-1 were 567.3, 11.9, 2.9 and 173.6 pg/mL, respectively.

To evaluate the relationship between galectin-3 and clinical and laboratory findings in patients undergoing maintenance hemodialysis, we further divided the patients into two subgroups based on the median value of plasma galectin-3. The high galectin-3 group had a higher proportion of older people with a lower MAP and was more likely to have lower FFMI and GNRI scores and be diagnosed with PEW than the low galectin-3 group. Additionally, patients in the high galectin-3 group were more likely to dialyze through a tunneled catheter and less likely to have an AV fistula than those in the low galectin-3 group. The high galectin-3 group also had significantly lower serum albumin levels but significantly higher circulating concentrations of IL-6, MCP-1, and TNF-α than the other group.

### 3.2. Correlation Analysis between Plasma Galectin-3 Levels, Body Composition, Circulating Inflammatory Markers, and Gnri in Hemodialysis Patients

The relationship between plasma galectin-3, systemic inflammatory markers, and various nutritional parameters, such as BMI, FFMI, and fat mass index (FMI), was determined using Spearman’s rank-order correlation. Galectin-3 positively correlated with the plasma levels of IL-6, TNF-α, and MCP-1 and negatively correlated with serum albumin, GNRI, and FFMI (shown in Figure 1). The three inflammatory markers (IL-6, TNF-α, and MCP-1) significantly correlated with each other and displayed similar trends regarding correlations with FFMI, GNRI, and serum albumin levels. Moreover, galectin-3 and the abovementioned inflammatory markers were not associated with FMI nor BMI in these individuals.

### 3.3. Factors Associated with Increased Plasma Galectin-3 Levels in Hemodialysis Patients

The factors influencing plasma galectin-3 levels in these patients were identified by logistic regression analysis (Table 2). In the univariate analysis, increasing age, a lower MAP, dialyzing with a catheter, presence of PEW (based on ISRNM criteria), and elevated plasma IL-6, TNF-α, and MCP-1 levels were associated with higher circulating concentrations of galectin-3. After adjustment for age, sex, and variables with *p* < 0.1 in the univariate analysis, MAP, tunneled cuffed catheters, and circulating TNF-α and MCP-1 remained significantly correlated with plasma galectin-3 concentrations.

### 3.4. Factors Associated with the Presence of PEW in Hemodialysis Patients

We next examined whether elevated galectin-3 levels could discriminate between hemodialysis patients with and without malnutrition. The ISRNM Diagnostic Criteria for PEW were used for the nutritional status assessment. Indeed, patients with PEW demonstrated significantly higher plasma galectin-3 levels than those without PEW (shown in Figure 2A). Furthermore, the diagnostic performance of the different inflammatory and nutritional markers for the presence of PEW was evaluated and compared using the ROC curves (shown in Figure 2B and Table 3). While GNRI and IL-6 achieved AUC values of 0.90 and 0.75, galectin-3 only achieved an AUC of 0.63.

In addition, the predictive factors of PEW in these patients were determined by logistic regression analyses (Table 4). Although plasma galectin-3 levels were significantly associated with PEW in unadjusted analyses, this association disappeared when several potential confounders were included in the multivariate model. Collectively, plasma galectin-3 concentrations were significantly elevated in chronic hemodialysis patients with PEW but did not show superiority in predicting the presence of PEW compared with other commonly used nutritional indicators.

## 4. Discussion

This study revealed that galectin-3, an important modulator of many biological processes, was significantly upregulated in the plasma of hemodialysis patients with elevated levels of inflammatory markers. In the ROC analysis, the diagnostic accuracy of circulating galectin-3 in predicting PEW was inferior to that of GNRI or IL-6. Additionally, although the plasma galectin-3 levels correlated with PEW severity, this correlation disappeared after adjustment for potential confounding variables. Therefore, this study does not support using plasma galectin-3 for routine nutritional assessment of these patient groups.

Low-grade systemic inflammation is a common hallmark of CKD and has been consistently associated with higher risks for death and severe complications [7]. Inflammation in ESRD may also impair the structural and functional reserves in different tissues, consequently disrupting the normal inter-organ crosstalk following acute stress and resulting in metabolic dyshomeostasis. Galectin-3 is expressed broadly in a set of immune cells, thereby implicated in the diverse inflammatory process of many diseases, such as CKD [34]. Several studies, including the present one, found that plasma galectin-3 positively correlates with systemic inflammatory markers in patients with CKD [18,35,36]. Our study also showed that tunneled dialysis catheters were associated with elevated levels of plasma galectin-3 compared to AV fistulas and grafts. Hemodialysis catheters have been known to cause chronic inflammation, irrespective of concurrent bacterial infection, which may explain in part this association. Moreover, a higher plasma galectin-3 concentration was considered as an unfavorable prognostic factor for all-cause mortality, cardiovascular events, and infectious diseases of these patients [23]. Hence, galectin-3 has the potential to improve risk discrimination in patients with impaired renal function. However, its performance still requires validation in large clinical trials.

Our results showed that the plasma galectin-3 level significantly increased in hemodialysis patients at risk of PEW, although it was not an independent predictor for PEW. This finding contradicts the results of studies conducted in patients with gastrointestinal malignancy [25,26], and the reason for this discrepancy remains unclear. Inadequate energy and protein intake is the most important determinant of PEW in patients with ESRD [37]. When the appetite of these individuals is decreased, the circulating levels of proinflammatory cytokines and the risk of adverse clinical outcomes are increased [38]. In the current study, the galectin-3 levels were inversely associated with FFMI, GNRI, and serum albumin level. However, we found that plasma galectin-3 and nPCR, which is a surrogate for daily protein intake, were not related, thereby possibly affecting the ability of galectin-3 to predict PEW.

Moreover, as noted above, an elevated plasma galectin-3 level is clearly associated with chronic inflammation in hemodialysis patients; however, it may not adequately reflect the severity of muscle and fat wasting at the patient level. Chronic inflammation can cause skeletal muscle atrophy partly via inducing local and systemic insulin resistance [39,40]. Galectin-3 interacts directly with the insulin receptor and impairs the downstream signaling pathway in insulin-target tissues; thus, this protein may link inflammation to reduced insulin sensitivity in affected individuals [41]. Meanwhile, endogenous galectin-3 expression, which is required for myogenic differentiation, enhances the efficiency of myogenesis in a mouse model of muscular dystrophy [42]. In summary, distinct pathophysiological pathways and molecular mechanisms of galectin-3 in body fat or skeletal muscle mass may explain, at least in part, inconsistencies between the published results. Therefore, the latest updated version of the KDOQI nutrition guidelines in CKD states that such inflammatory biomarkers should not be used in isolation but rather in combination with other comprehensive evaluation approaches when assessing the nutritional status of adults with CKD [43].

However, this study has several limitations. First, because of small sample size of this study, future studies must have a sufficiently large sample size to confirm these findings. Second, the study has a cross-sectional study design, which cannot determine causal effects or analyze the long-term changes in nutritional status. Hence, the research results must be interpreted cautiously. Third, unmeasured confounding effects may still exist despite adjusting for potential confounders by multiple regression. Fourth, impaired renal clearance, cardiac inflammation, activated platelets, and other etiologies may underlie the elevated circulating levels of galectin-3 in hemodialysis patients [44,45,46]. A close relationship between activation of the renin-angiotensin-aldosterone system and galectinse has also been observed in previous studies [46]. However, we did not have samples from urine and dialysate to evaluate the clearance rate of galectin-3, and no measurements of tissue concentrations of galectin-3 were available. Thus, the major source of circulating galectin-3 in patients with ESRD cannot be determined from this study. Further studies are needed to clarify this limitation. Finally, lean tissue measurements by BIA are influenced by hydration status. Therefore, special care should be taken when interpreting these results in subjects with edema.

## 5. Conclusions

Inflammation and PEW are common findings in patients undergoing maintenance hemodialysis. In these patients, the level of plasma galectin-3 is associated with an increased risk for chronic systemic inflammation but not PEW. In addition, galectin-3 did not show superiority in predicting PEW compared with GNRI, which is a well-validated nutritional screening tool for patients at risk for PEW. In future studies, identification of novel biomarkers is required to detect patients at risk for malnutrition and implement appropriate interventions to decrease the burden of CKD.

## Figures and Tables

**Figure 1 nutrients-13-02803-f001:**
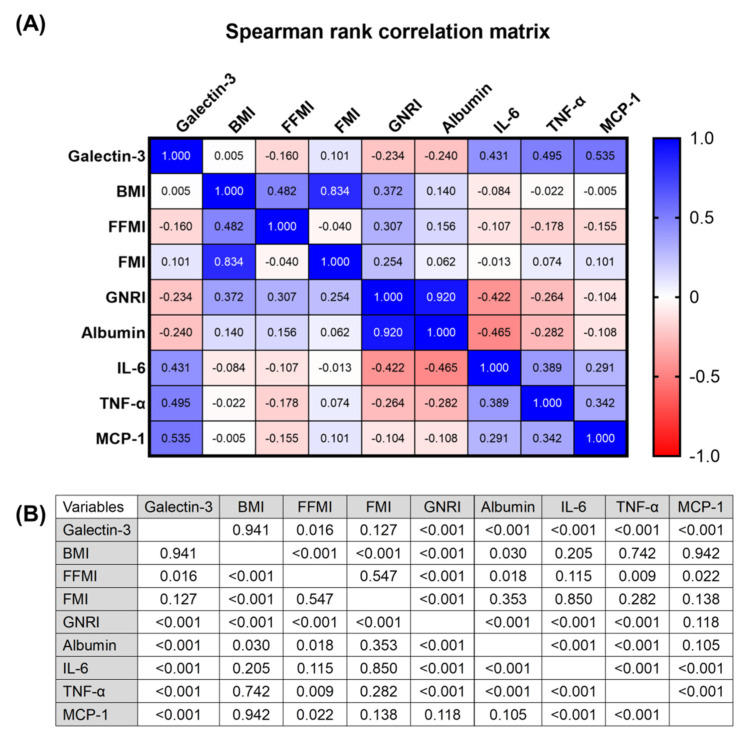
(**A**) The correlation matrix summarizes the strength of the association among body composition indices, systemic inflammatory factors, geriatric nutritional risk index, and plasma galectin-3 in adults undergoing maintenance hemodialysis. The numbers in each cell represent the value of the Spearman’s correlation coefficient. (**B**) The table represents the corresponding *p* values in (**A**). BMI: body mass index, FFMI: fat-free mass index, FMI: fat mass index, GNRI: geriatric nutritional risk index,IL-6:interleukin-6, TNF-α: tumor necrosis factor-alpha, MCP-1: monocyte chemoattractant protein-1

**Figure 2 nutrients-13-02803-f002:**
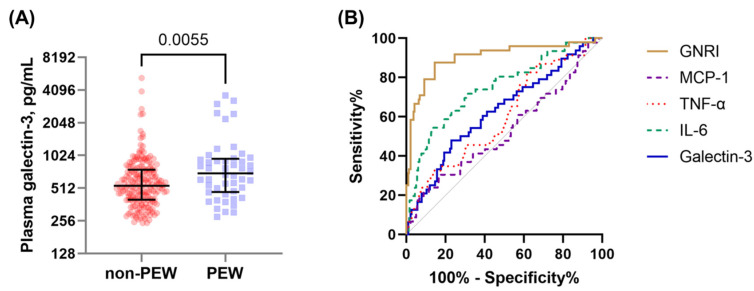
Plasma galectin-3 concentrations and predictive power for PEW in chronic hemodialysis patients. (**A**) In hemodialysis patients with PEW, plasma galectin-3 concentrations were significantly (*p* < 0.01) elevated as compared to subjects without PEW. (**B**) Receiver operating characteristic (ROC) curve analyses comparing the predictive accuracy in identifying PEW of galectin-3 with other classical nutritional and inflammatory indices.

**Table 1 nutrients-13-02803-t001:** Demographic, clinical characteristics and laboratory findings of maintenance hemodialysis patients stratified by the median plasma levels of galectin-3.

Parameters ^a^	All Patients (n = 240)	Galectin-3 ≤ 567 pg/mL (n = 120)	Galectin-3 > 567 pg/mL (n = 120)	*p* Value
Galectin-3, pg/mL	567.3 (406.5−808.5)	406.9 (332.0–480.2)	806.3 (657.5–987.0)	<0.001
**Demographic and clinical characteristics**				
Age, years	64.5 (55.3–74.0)	61.0 (52.5–69.0)	68.5 (58.3–76.0)	0.001
Women, *n* (%)	86 (35.8)	41 (34.2)	45 (37.5)	0.590
Dialysis vintage, months	49.0 (29.0–93.8)	48.5 (28.0–84.8)	49.0 (29.0–108.3)	0.740
spKt/V_urea_	1.53 ± 0.26	1.53 ± 0.25	1.53 ± 0.26	0.983
nPCR, g/kg/day	1.13 (1.00–1.30)	1.11 (1.00–1.26)	1.15 (0.98–1.35)	0.258
Hypertension, *n* (%)	212 (88.3)	108 (90.0)	104 (86.7)	0.421
Diabetes mellitus, *n* (%)	134 (55.8)	62 (51.7)	72 (60.0)	0.194
Cardiovascular disease, *n* (%)	113 (47.1)	52 (43.3)	61 (50.8)	0.244
Mean arterial pressure, mmHg	101 ± 18	105 ± 17	97 ± 18	0.001
**Dialyzer**				0.776
High-flux, *n* (%)	170 (70.8)	84 (70.0)	86 (71.7)	
Low-flux, *n* (%)	70 (29.2)	36 (30.0)	34 (28.3)	
**Vascular access**				0.001
AV fistula, *n* (%)	188 (78.3)	102 (85.0)	86 (71.7)	
AV graft, *n* (%)	28 (11.7)	15 (12.5)	13 (10.8)	
Tunneled catheter, *n* (%)	24 (10.0)	3 (2.5)	21 (17.5)	
**Anthropometric measurements and nutritional scores**				
Body mass index, kg/m^2^	24.3 (21.9–27.3)	24.0 (22.2–27.0)	24.9 (21.3–27.6)	0.642
Fat-free mass index, kg/m^2^	16.4 (14.7–18.1)	16.8 (15.1–18.4)	15.9 (14.5–17.7)	0.026
Fat mass index, kg/m^2^	8.1 (5.2–10.6)	7.5 (5.0–10.2)	8.8 (5.3–10.7)	0.069
ECW/TBW, %	40.1 (39.2–41.2)	40.0 (39.1–40.9)	40.4 (39.3–41.5)	0.107
GNRI	101.3 (97.2–105.7)	102.7 (98.3–106.7)	100.3 (95.7–102.7)	0.003
GNRI ≤ 98, *n* (%)	62 (25.8)	25 (20.8)	37 (30.8)	0.077
Presence of PEW ^b^, *n* (%)	48 (20.0)	17 (14.2)	31 (25.8)	0.024
**Laboratory test results**				
Hemoglobin, g/dL	9.8 (9.0–10.7)	9.9 (9.0–10.7)	9.8 (9.1–10.7)	0.993
Leuocytes, 10^3^/μL	5.7 (4.7–7.0)	5.9 (5.0–7.0)	5.4 (4.6–7.1)	0.232
Albumin, g/dL	4.1 (3.9–4.3)	4.1 (3.9–4.4)	4.0 (3.8–4.1)	0.002
Calcium, mg/dL	9.1 (8.5–9.6)	9.2 (8.7–9.7)	9.0 (8.5–9.6)	0.252
Phosphorous, mg/dL	5.1 (4.3–6.0)	5.1 (4.3–5.9)	5.1 (4.3–6.1)	0.999
Total cholesterol, mg/dL	142 (124–167)	145 (125–173)	138 (123–165)	0.099
Glucose, mg/dL	123 (99–165)	114 (94–170)	131 (102–165)	0.070
**Plasma biomarkers**				
IL-6, pg/mL	11.9 (6.9–22.8)	8.8 (3.8–15.5)	16.0 (9.6–32.3)	<0.001
TNF-α, pg/mL	2.9 (1.2–5.1)	1.3 (1.0–3.9)	4.0 (2.4–6.1)	<0.001
MCP-1, pg/mL	173.6 (129.2–231.8)	139.1 (102.6–180.3)	208.9 (167.6–271.8)	<0.001

Abbreviations: AV, arteriovenous; ECW/TBW, the ratio of extracellular water to total body water; GNRI, geriatric nutritional risk index; IL-6, interleukin-6; MCP-1, monocyte chemoattractant protein-1; nPCR, normalized protein catabolic rate; PEW, protein-energy wasting; spKt/V_urea_, single pool Kt/V_urea_; TNF-α, tumor necrosis factor-alpha. ^a^ The results are expressed as *n* (%) for the categorical data and as mean ± SD or median and interquartile range (IQR) for the continuous normally or non-normally distributed variables. ^b^ Protein-energy wasting was defined according to the diagnostic criteria of the International Society of Renal Nutrition and Metabolism (ISRNM).

**Table 2 nutrients-13-02803-t002:** Independent predictors of high plasma levels of galectin-3 in chronic hemodialysis patients.

	High Plasma Level of Galectin-3			
	Univariate		Multivariate ^a^	
Variables	Odds Ratio (95% CI)	*p* Value	Odds Ratio (95% CI)	*p* Value
Age (per year)	1.034 (1.015–1.054)	0.001	1.009 (0.984–1.035)	0.483
Gender (male:female)	0.865 (0.510–1.467)	0.590	1.269 (0.612–2.634)	0.522
Dialysis vintage (per year)	1.015 (0.961–1.073)	0.589	–	–
spKt/V_urea_ (per 0.1 unit)	1.001 (0.906–1.106)	0.983	–	–
Hypertension (yes:no)	0.722 (0.326–1.600)	0.423	–	–
Diabetes mellitus (yes:no)	1.403 (0.841–2.340)	0.194	–	–
Cardiovascular disease (yes:no)	1.352 (0.813–2.248)	0.245	–	–
MAP (per 10 mmHg)	0.773 (0.663–0.901)	0.001	0.761 (0.614–0.943)	0.013
High-flux (yes:no)	1.084 (0.621–1.892)	0.776	–	–
Tunnel catheter (yes:no)	8.273 (2.396–28.558)	0.001	7.313 (1.396–38.318)	0.019
ECW/TBW, %	1.076 (0.918–1.261)	0.366	–	–
Presence of PEW (yes:no)	2.110 (1.095–4.067)	0.026	1.231 (0.476–3.180)	0.668
Hemoglobin (per g/dL)	1.093 (0.896–1.333)	0.381	–	–
Leuocytes (per 10^3^/μL)	0.922 (0.815–1.043)	0.196	–	–
Calcium (per mg/dL)	0.856 (0.632–1.159)	0.313	–	–
Phosphorous (per mg/dL)	1.020 (0.861–1.207)	0.820	–	–
Total cholesterol (per mg/dL)	0.994 (0.987–1.001)	0.079	1.000 (0.990–1.010)	0.989
Glucose (per mg/dL)	1.003 (0.999–1.006)	0.166	–	–
IL-6 (per pg/mL)	1.034 (1.014–1.055)	0.001	1.020 (0.999–1.041)	0.064
TNF-α (per pg/mL)	1.459 (1.267–1.681)	<0.001	1.374 (1.167–1.617)	<0.001
MCP-1 (per pg/mL)	1.012 (1.008–1.017)	<0.001	1.009 (1.004–1.014)	<0.001

Abbreviations: CI, confidence interval; ECW/TBW, the ratio of extracellular water to total body water; IL-6, interleukin-6; MAP, mean arterial pressure; MCP-1, monocyte chemoattractant protein-1; PEW, protein-energy wasting; spKt/V_urea_, single pool Kt/V_urea_; TNF-α, tumor necrosis factor-alpha. ^a^ The multivariable model is adjusted for model including age, gender, MAP, dialyzing with a catheter, presence of PEW, total cholesterol, IL-6, TNF-α, and MCP-1. The VIF for each variable in the multivariate model was less than 2.5.

**Table 3 nutrients-13-02803-t003:** Receiver operating characteristic curve analyses of the diagnostic performance of inflammatory and nutritional markers in identifying the presence of PEW in maintenance hemodialysis patients.

	AUC (95% CI)	*p*-Value	SEN ^a^ (%)	SPEC ^a^ (%)	PPV ^a^ (%)	NPV ^a^ (%)	PLR ^a^	NLR ^a^
GNRI	0.90 (0.86–0.94) ^b^	<0.0001	87.5	83.3	56.8	96.4	5.25	0.15
IL-6	0.75 (0.69–0.80) ^c^	<0.0001	54.4	86.3	50.0	88.2	3.96	0.53
Gal-3	0.63 (0.56–0.69) ^d^	0.0057	47.9	77.1	34.3	85.5	2.09	0.68
TNF-α	0.60 (0.54–0.67)	0.0270	82.6	38.5	25.3	89.7	1.34	0.45
MCP-1	0.53 (0.46–0.59)	0.5997	30.4	83.5	31.8	82.6	1.85	0.83

Abbreviations: AUC, area under curve; CI, confidence interval; Gal-3, galectin-3; GNRI, geriatric nutritional risk index; IL-6, interleukin 6; MCP-1, monocyte chemoattractant protein-1; NLR, negative likelihood ratio; NPV, negative predictive value; PLR, positive likelihood ratio; PPV, positive predictive value; SEN, sensitivity; SPEC, specificity; TNF-α, tumor necrosis factor-alpha. ^a^ Values were calculated by the Youden’s index that maximized the sum of sensitivity and specificity. ^b^ The AUC of GNRI was significantly greater than the AUC values of IL-6, galectin-3, TNF-α, and MCP-1 for predicting PEW (*p* = 0.0017, <0.0001, <0.0001, and <0.0001, respectively). ^c^ The AUC of IL-6 was significantly greater than the AUC values of galectin-3, TNF-α, and MCP-1 for predicting PEW (*p* = 0.0109, 0.0059, and <0.0001, respectively). ^d^ The AUC of galectin-3 was significantly greater than that of MCP-1 for predicting PEW (*p* = 0.0171).

**Table 4 nutrients-13-02803-t004:** Logistic regression analyses of independent variables associated with the presence of PEW as defined by ISRNM criteria in chronic hemodialysis patients.

Variables ^a^	Model 1 ^b^		Model 2 ^c^		Model 3 ^d^		Model 4 ^e^	
	Odds Ratio (95% CI)	*p* Value	Odds Ratio (95% CI)	*p* Value	Odds Ratio (95% CI)	*p* Value	Odds Ratio (95% CI)	*p* Value
Galectin-3 (per pg/mL)	1.001 (1.000–1.001)	0.013	1.000 (1.000–1.001)	0.058	1.000 (0.999–1.001)	0.888	1.000 (0.999–1.001)	0.630
Age (per year)			1.032 (1.007–1.059)	0.013	1.014 (0.986–1.041)	0.329	0.984 (0.948–1.021)	0.388
Gender (male:female)			0.709 (0.365–1.378)	0.311	0.657 (0.289–1.494)	0.316	1.003 (0.225–4.462)	0.997
Tunnel catheter (yes:no)					1.876 (0.433–8.119)	0.400	1.038 (0.084–12.770)	0.977
Hemoglobin (per g/dL)					0.718 (0.510–1.012)	0.058	0.906 (0.582–1.410)	0.662
Calcium (per mg/dL)					0.671 (0.419–1.074)	0.097	1.268 (0.689–2.334)	0.446
Phosphorous (per mg/dL)					0.685 (0.512–0.916)	0.011	0.952 (0.615–1.473)	0.825
Total cholesterol (per mg/dL)					0.977 (0.964–0.991)	0.001	0.978 (0.959–0.997)	0.025
MCP-1 (per pg/mL)					1.000 (0.998–1.003)	0.680	1.001 (0.997–1.004)	0.724
nPCR (per g/kg/day)							0.666 (0.507–0.875)	0.003
GNRI (per 1 unit)							0.691 (0.595–0.804)	< 0.001
FFMI (per 1 unit)							0.912 (0.667–1.246)	0.563
ECW/TBW, %							1.018 (0.774–1.340)	0.896

Abbreviations: CI, confidence interval; ECW/TBW, the ratio of extracellular water to total body water; FFMI, fat-free mass index; GNRI, geriatric nutritional risk index; MCP-1, monocyte chemoattractant protein-1; nPCR, normalized protein catabolic rate; PEW, protein-energy wasting. ^a^ The *p* value of the variables listed in this Table was less than 0.1 in univariate analysis. Additionally, there was no multicollinearity in the multiple regression model because the VIF for each parameter was less than 2.5. ^b^ Univariate analysis. ^c^ Adjusted for galectin-3, age, and gender. ^d^ Adjusted for galectin-3, age, gender, dialyzing with a catheter, hemoglobin, calcium, phosphate, total cholesterol, and MCP-1. ^e^ Adjusted for galectin-3, age, gender, dialyzing with a catheter, hemoglobin, calcium, phosphate, total cholesterol, MCP-1, nPCR, GNRI, FFMI, and ECW/TBW.

## Data Availability

The data analyzed in this study are not publicly available because individual privacy may be compromised. Interested groups could contact Ming-Tsun Tsai at mingtsun74@gmail.com to request permission to access these datasets.

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
