# Peer review of "Relationship between Circulating Galectin-3, Systemic Inflammation, and Protein-Energy Wasting in Chronic Hemodialysis Patients"

_nutrients, 2021, doi:10.3390/nu13082803_

Round 1
Reviewer 1 Report
General comments
The present cross-sectional study measured serum galectin-3 levels in patients receiving maintenance hemodialysis and determined the associations between the serum galectin-3 level and serum inflammatory markers and other nutritional surrogate markers. They found that serum galectin-3 levels were increased in hemodialysis patients and were positively associated with increased serum inflammatory markers, whereas negatively associated with serum albumin level. Serum galectin-3 levels were increased in patients with protein-energy wasting (PEW). The authors also compared the predictability for PEW in those patients and showed that serum galectin-3 was inferior to GNRI and serum TNF-a levels when ROCs were compared among them. They also determined the factors that were associated with PEW defined by the ISNRM and revealed that serum galectin-3 level was not associated with the PEW criteria in the multivariable-adjusted model.
The paper is well-written and met most of the requirements that the journal requested and the general observational study demanded. There are several points to be addressed before publication.
Specific comments
Major
#1. In Table 4, GNRI was included as a covariate in the multivariable model. Because the criteria of the PEW included both serum albumin and BMIN, it is quite natural that GNRI was strongly correlated with PEW. The association of a surrogate marker with PEW is largely dependent on the definition of PEW. Namely, which nutritional parameters are included to diagnose PEW. In this regard, it would be very interesting if the correlations between the serum inflammatory markers including serum galectin-3 levels and lean body mass measured by multifrequency bioelectrical impedance analysis were compared. Skeletal muscle mass is another surrogate marker of nutritional status. The results may be different when other definition of PEW would be applied.
#2. The authors described the association between the serum galectin-3 levels and the PEW became non-significant after multivariable adjustments. Which covariate had the greatest impact on the attenuated association between the serum galectin-3 levels and the PEW? Namely, after which factor was incorporated did the association become non-significant? GNRI or other factors? The factor may be the confounding factor. OR the factor may be closely correlated with the serum galectin-3 level.
#3. The mechanisms by which galectin-3 levels were increased in hemodialysis patients should be described in the Introduction section or Discussion section for the readers. The factors regulating serum galectin-3 levels should be also mentioned, if such previous studies are present.
#4. Why did the author choose “0.25” as the cut-off values for incorporating covariate in the multivariable model? Generally, “0.1” is used to incorporate variables in the univariate analysis into the multivariable model?
#5. In Tables 2 and 4, MCP, IL-6, and galectin-3 were incorporated in the multivariable model. These three markers showed significant correlation with each other, and multi-co-linearity may be problematic when all these three markers were used in the same model. The non-significant association may be altered when serum levels of MCP-1 and TNF-a were excluded in the multivariable model for PEW.
Author Response
Comments and Suggestions of reviewer 1
- In Table 4, GNRI was included as a covariate in the multivariable model. Because the criteria of the PEW included both serum albumin and BMI, it is quite natural that GNRI was strongly correlated with PEW. The association of a surrogate marker with PEW is largely dependent on the definition of PEW. Namely, which nutritional parameters are included to diagnose PEW. In this regard, it would be very interesting if the correlations between the serum inflammatory markers including serum galectin-3 levels and lean body mass measured by multifrequency bioelectrical impedance analysis were compared. Skeletal muscle mass is another surrogate marker of nutritional status. The results may be different when other definition of PEW would be applied.
Response: Thanks for your suggestions. In the univariate analysis, the fat free mass index (FFMI), a surrogate of lean body mass, was inversely associated with the presence of PEW (OR, 0.766; 95% CI, 0.649–0.903; P = 0.002). Therefore, we included FFMI in the multivariable model to determine the independent predictor for the presence of PEW. Our results revealed that FFMI was not an independent predictor of PEW after multivariate analysis. These findings were presented in the revised Table 4 (page 10, Table 4).
- The authors described the association between the serum galectin-3 levels and the PEW became non-significant after multivariable adjustments. Which covariate had the greatest impact on the attenuated association between the serum galectin-3 levels and the PEW? Namely, after which factor was incorporated did the association become non-significant? GNRI or other factors? The factor may be the confounding factor. OR the factor may be closely correlated with the serum galectin-3 level.
Response: Thanks for your suggestions. After adjustment for age and sex, the association between the plasma galectin-3 levels and the PEW became non-significant. These findings were presented in the revised Table 4 (page 10, Table 4).
- The mechanisms by which galectin-3 levels were increased in hemodialysis patients should be described in the Introduction section or Discussion section for the readers. The factors regulating serum galectin-3 levels should be also mentioned, if such previous studies are present.
Response: Thanks for your suggestions. We have added the information about increased galectin-3 levels in hemodialysis patients in the revised manuscript as follows: “Fourth, impaired renal clearance, cardiac inflammation, activated platelets, and other etiologies may underlie the elevated circulating levels of galectin-3 in hemodialysis patients [44-46]. A close relationship between activation of the renin-angiotensin-aldosterone system and galectin‐3 has also been observed in previous studies [46]. However, we did not have samples from urine and dialysate to evaluate the clearance rate of galectin-3, and no measurements of tissue concentrations of galectin-3 were available. Thus, the major source of circulating galectin-3 in patients with ESRD cannot be determined from this study. Further studies are needed to clarify this limitation.” (page 12, Para 1, Line 1-8)
- Why did the author choose “0.25” as the cut-off values for incorporating covariate in the multivariable model? Generally, “0.1” is used to incorporate variables in the univariate analysis into the multivariable model?
Response: Thanks for your suggestions. We have adapted the P value of < 0.1 as the cutoff to select variables in the univariate analysis into the multivariable model. These findings were presented in the revised Table 2 and 4 (page 4, Para 3, Line 8-10; page 7, Para 1, Line 5; page 7, Table 2; page 10, Table 4).
- In Tables 2 and 4, MCP, IL-6, and galectin-3 were incorporated in the multivariable model. These three markers showed significant correlation with each other, and multi-co-linearity may be problematic when all these three markers were used in the same model. The non-significant association may be altered when serum levels of MCP-1 and TNF-a were excluded in the multivariable model for PEW.
Response: Thanks for your suggestions. Multicolinearity was identified by calculating the variance inflation factor (VIF) for each predictor in multivariable models, and a VIF ≥ 2.5 indicated considerable collinearity. In this study, the VIF for each variable in Table 2 and 4 was less than 2.5. Therefore, there was no significant collinearity between these variables (page 4, Para 3, Line 10-12; page 8, the footnote of Table 2; page 10, the footnote of Table 4).

Reviewer 2 Report
The authors investigated the value of galectin-3 as a possible marker of protein-energy wasting (PEW) in 240 chronic hemodialysis patients. Galectin 3 High Galectin-3 was correlated with low mean arterial pressure and markers of inflammation. Although plasma Galectin-3 levels correlated with the severity of PEW, this correlation disappeared after multivariable adjustment.
Major: None
Minor:
- Line 91 ff: “(1) with total hemodialysis treatment time < 12 hours per week….”
I suggest to omit the “with” here and in the following sentences up to line 93 - Line 100: Why is Diabetes defined only as treated Diabetes and not also as a fasting blood glucose level > 126 mg/dl?
- Line 116: “…, we detected biochemical parameters…”. I suggest to replace detected by measured.
- Line 136: GNRI has been explained once in the abstract section but this is the first time it appears in the main body of this manuscript and should therefore be explained (geriatric nutritional risk index)
Author Response
Comments and Suggestions of reviewer 2
- Line 91 ff: “(1) with total hemodialysis treatment time < 12 hours per week….” I suggest to omit the “with” here and in the following sentences up to line 93.
Response: Thanks for your suggestions. We have omitted the “with” in the sentences from Line 91 to line 93. The sentences are revised accordingly with “(1) total hemodialysis treatment time < 12 hours per week (n = 6); (2) having a prosthetic limb (n = 2); (3) having an implanted pacemaker or cardioverter-defibrillator (n = 2); and (4) having a history of solid tumors (n = 3), bacterial infections (n = 5), or hepatobiliary disease (n = 4).” (page 2, Para 4, Line 4-7)
- Line 100: Why is Diabetes defined only as treated Diabetes and not also as a fasting blood glucose level > 126 mg/dl?
Response: Thanks for your suggestions. This study was limited by its retrospective nature, and diabetes mellitus can only be defined by using the available clinical data but not by the diagnostic criteria from the American Diabetes Association. The revised sentences were as follows: “The complete medical history of each patient was recorded and retrospectively assessed.” (page 3, Para 2, Line 1-2)
- Line 116: “…, we detected biochemical parameters…”. I suggest to replace detected by measured.
Response: Thanks for your suggestions. We have modified the sentence as follows: “we measured biochemical parameters such as serum albumin, blood urea nitrogen, creatinine, calcium, and phosphate.” (page 3, Para 4, Line 1-3)
- Line 136: GNRI has been explained once in the abstract section but this is the first time it appears in the main body of this manuscript and should therefore be explained (geriatric nutritional risk index)
Response: Thanks for your suggestions. We have modified the sentence as follows: “Malnutrition was identified by the Geriatric Nutrition Risk Index (GNRI) and the International Society of Renal Nutrition and Metabolism (ISRNM) criteria for PEW [31,32].” (page 3, Para 6, Line 1-2)

Round 2
Reviewer 1 Report
The authors have responded to the queries raised by the reviewer appropriately.
Author Response
Response: Thank you very much.